# TrendDiff: Decoupling Intrinsic and Measurement Trends for Enhanced Time Series Causal Discovery

## Abstract

Time trends can be classified into intrinsic (real) and measurement (false) trends. There has long been a critical need for techniques to discern them, especially in investment decision-making. In causal discovery, these measurement trends, essentially measurement errors, can significantly impact the performance of algorithms, making it crucial to identify and eliminate them before analysis as well. Recognizing this need, we present a novel algorithm, termed Trend Differentiator (TrendDiff). It is capable of detecting all trend-influenced variables and differentiating between those affected by measurement trends and those displaying intrinsic trends, relying on changing causal module detection and trend-influenced variables' structural properties, respectively. Extensive experiments on synthetic and real-world data demonstrate the efficacy of this approach.

## 1 Introduction

Emerging in the early 1990s, causal discovery algorithms have undergone substantial growth in the past decades (Spirtes and Zhang, 2016). These algorithms strive to infer causality from purely observational data, serving as valuable instruments in situations where randomized controlled trials are rendered impractical due to ethical concerns, financial constraints, and other obstacles. Standing at the intersection of explosive data volumes and advancements in computational capabilities, a surge in theoretical and applied causal research has ensued. However, the rapid accumulation of data presents not only exciting possibilities but significant challenges in causal discovery.

A prevalent challenge is the presence of time trends, frequently encountered in time series. As articulated by Phillips (2005), "No one understands trends, but everyone sees them in data". Prior research has extensively investigated the impact of trends on the efficacy of conventional statistical algorithms (White and Granger, 2011; Wu et al., 2007). Yet their influence on causal discovery remains largely unexplored. Based on the origin, trends can be classified into two distinct categories: intrinsic (real) and measurement (false) trends. In this context, we define the terms "trend", "intrinsic trend", and "measurement trend" as follows:

**Definition 1.** *A **trend** is a function concerning time within a given data span. Specificly, time trend $T = f(t)$, for $t_{start} \leq t \leq t_{end}$.*

**Definition 2.** *An **intrinsic trend** is inherent to the fundamental mechanisms governing the variables (e.g., global warming, the temperature is really increasing).*

**Definition 3.** *A **measurement trend** is essentially an observation error unique to the recorded values (e.g., an observed increase in diagnosed thyroid nodule patients due to enhanced medical techniques, despite a stable real incidence rate over time, see **Figure 1**).*

The two types of trends originate from distinct sources, exert disparate impacts, and necessitate differential treatment.

However, there is this impression – time trends, be it an intrinsic trend, or a measurement trend, should be removed before analyses – which is not accurate. Undoubtedly, measurement trends, being a form of measurement error, necessitate removal. Take constraint-based causal discovery methods, which rely on conditional independence tests, for example. **Figure 2 (a)** shows the true causal graph,

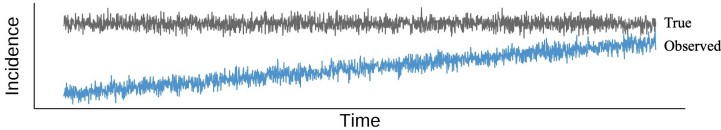

Figure 1: The true and observed incidence of the thyroid nodule along time – a typical example of measurement trends.

the variables $\underline{X_2}$ and $\underline{X_3}$ are not observable, with a measurement trend exhibited in the observed $X_2$ and $X_3$. These measurement trends greatly increase the noise in these variables. For $X_2$, this rise in noise alters its relationship with its neighbors $X_1$ and $X_3$, weakening the observed dependencies as the measurement trends intensify. Additionally, the inability to accurately observe $X_2$'s true values hampers its capacity to d-separate $X_1$ and $X_3$, due to the challenge in precisely controlling for $X_2$. Analogous phenomena transpire for another measurement-trend variable $X_3$. The causal network identified in **Figure 2 (b)** diverges significantly from the ground truth in such scenarios. To summarize, measurement trends, inherently measurement errors, introduce two issues for constraint-based causal discovery: 1. the dependence between measurement-trend variables and their neighbors weakens with increasing trends; 2. the conditional independence given the measurement-trend variables vanishes, yielding increasing dependence (Scheines and Ramsey, 2016; Zhang et al., 2017). As highlighted in earlier research regarding measurement error in causal discovery, this influence is not limited to constraint-based causal algorithms but also extends to other methodologies, including those based on functional causal models (Zhang et al., 2017). Conversely, intrinsic trends are integral components of the variables and mechanisms, facilitating the identification of underlying causal relationships. Removal of intrinsic trends would decrease the signal-to-noise ratio, leading to lower detection power, and thus should be avoided. Consequently, discerning between intrinsic and measurement trends is crucial before conducting causal discovery analyses.

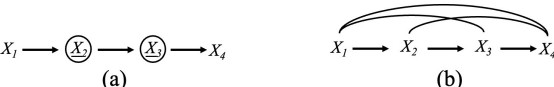

Figure 2: An illustration of how ignoring measurement trends in causal discovery may lead to spurious connections by constraint-based methods. (a) The true causal graph. Variables whose actual values do not match the observed ones are underlined to indicate their true values. Encircled variables signify their unobservability. Here, the circled $\underline{X_2}$ and $\underline{X_3}$ represent the true, but unobservable, values of the measurement-trend variables. (b) The estimated skeleton based on observed data.

This study introduces the Trend Differentiator (TrendDiff) algorithm, designed to pinpoint variables influenced by trends and differentiate between those affected by measurement trends and those displaying intrinsic trends. It is not only critical in data pre-processing for causal discovery but carries substantial practical importance in decision-making. Discerning true market trends from transient fluctuations is essential for avoiding misallocation of resources in non-viable market opportunities. The ability to accurately identify trend types is key to reducing such investment risks.

The principal contributions of our work are shown below:

- **Problem Formulation**. We parameterize variables with intrinsic and measurement trends using graphical models. While there has already been research regarding measurement errors in causal discovery, no attention has been paid to differentiating time trends. However, as we motivated above, distinguishing intrinsic from measurement trends is of great theoretical and practical value. To the best of our knowledge, this work is the first to formally propose this problem.
- **TrendDiff Algorithm.** Employing the method of detecting changing causal modules, we can efficiently identify all variables affected by trends. Subsequently, by harnessing the unique causal structures under intrinsic and measurement trends, we are able to distinguish between them. Integrating these technologies, we present the TrendDiff algorithm, a novel solution specifically designed for the discernment of time trends.
- **Experimental Validation.** We use extensive experimental evaluations, including analyses of a real-world dataset, to demonstrate the robustness and utility of our algorithm.

## 2 PARAMETERIZING TRENDS AND RELATED WORK

### 2.1 PARAMETERIZING TIME TRENDS

To put intrinsic and measurement trends clearer, we resort to structural equation models (SEMs), where each variable $V_i$ is formulated as a function of its direct causes and an error term $\varepsilon_i$. Here $\varepsilon_i$ encapsulates all other unmeasured causes of $V_i$, with the $\varepsilon_i$ values for different variables being mutually independent. **Figure 3 (a)** depicts a simple causal model, where a direct causal chain is established from variable $X_1$, leading to $X_2$, and subsequently to $X_3$. Each variable is associated with a structural equation, and the model can be parameterized by assigning exact functions to $f(V_i)$, as well as a joint normal distribution to $\varepsilon_1, \varepsilon_2, \varepsilon_3 \sim \mathcal{N}(\mu, \Sigma^2)$. In this case, $\Sigma^2$ is diagonal, reflecting the independence among the error terms $\varepsilon_1$, $\varepsilon_2$, and $\varepsilon_3$. Regardless of the functions and free parameter values assigned, the model in Figure 3 (a) exhibits conditional independence: $X_1 \perp\!\!\!\perp X_3 | X_2$. In **Figure 3 (b)**, we present the same model as in Figure 3 (a) but with an added intrinsic trend $T_2$ affecting $X_2$. The intrinsic trend $T_2$ impacts the generation of $X_2$ and is an inherent part of its underlying mechanisms. In this case, the observed and real values of $X_2$ are identical. The added intrinsic time trend can go into the causal network through $X_2$ without altering the original causal structure. Consequently, a trend in $X_3$ can be observed, which arises due to the influence of $T_2$. In **Figure 3 (c)**, we depict the same model but with true values $\underline{X_2}$ being "measured" as $X_2$, accompanied by a measurement trend $T_2$. In this case, the real and observed values of $X_2$ differ. The measurement trend $T_2$ is present only in the observed $X_2$. Due to the collider at $X_2$, $T_2$ cannot influence the real values $\underline{X_2}$ and is unable to propagate through the original causal network. To summarize, intrinsic and measurement trends are fundamentally the same in form (a function concerning time within a given data span). However, intrinsic trends affect the true value of variables, whereas measurement trends do not.

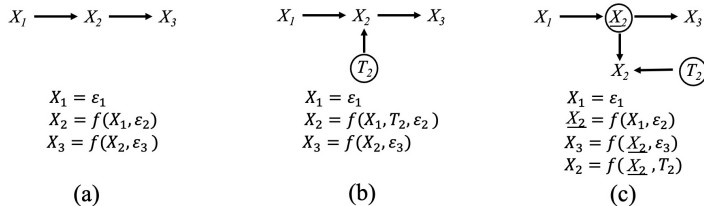

Figure 3: Causal models for variables with trends and corresponding equations. (a) A chain graph without trend. (b) $X_2$ with an intrinsic trend. (c) $X_2$ with a measurement trend.

### 2.2 RELATED WORK

**Measurement error in causal discovery.** Fundamentally, measurement trends represent a problem of measurement error, which adversely affects causal discovery performance. There has already been some research on measurement error in causal discovery. In linear Gaussian contexts, Scheines and Ramsey (2016) parameterized measurement error using SEMs and explored the effect of Gaussian measurement error on fast greedy equivalence search (FGES). Then identifiability conditions in linear Gaussian situations are discussed in Zhang et al. (2017) through factor analysis, with a key identified challenge being the unknown variances of measurement errors $\mathbf{E}$. If known, the covariance matrix of $\tilde{\mathbf{X}}$ would be easily accessible and readily used. To address this, Blom et al. (2018) offers an estimate for the upper bound of $\mathbf{E}$, while Saeed et al. (2020) proposes a consistent partial correlations estimator. In linear non-Gaussian scenarios, Zhang et al. (2018) demonstrates the identifiability of the *ordered group decomposition* of $\tilde{G}$, which contains crucial causal information. However, this method depends on over-complete independent component analysis (OICA (Hyvärinen and Oja, 2000)), hindered by issues of local optima and high computational complexity (Hoyer et al., 2008; Shimizu et al., 2009), making the practical application of Zhang et al. (2018)'s theoretically sound results challenging. Given this, Dai et al. (2022) defined the Transformed Independent Noise (TIN) condition and exploited it to identify the *ordered group decomposition* by independence tests.

Particularly regarding the differentiation of intrinsic and measurement trends, this study stands as the first to offer a solution. Distinct from the above-mentioned studies on causal discovery in the presence

of measurement error, our research uniquely: 1) distinguish the two types of trends, facilitating data preprocessing to significantly improve data quality. It can be integrated with various analytical tools; 2) extends its utility beyond merely enhancing causal discovery. It possesses direct and significant practical relevance in investment decision-making.

## 3 ASSUMPTIONS

Instead of relying on the conventional assumption of causal sufficiency, this research adopts a modified concept, termed "pseudo causal sufficiency" (Huang et al., 2020). Traditional causal sufficiency assumes that all common causes (confounders) influencing observed variables are captured in the dataset. However, the occurrence of time trends presents a challenge to this assumption. Time trends typically emerge from intricate, compounded factors, and time trends across various variables might be interlinked owing to certain hidden confounders. These confounders could represent high-level background factors, like economic policies in the stock market. Therefore, rather than assuming the absence of unobserved confounders, our approach operates under the assumption of pseudo causal sufficiency. This assumption signifies that the only unobserved confounders are those inherent in time trends.

**Assumption 1** (Pseudo causal sufficiency)**.** Any potential confounders can be encapsulated by a mathematically smooth time function. It follows that at each time instance, the values of these confounders are fixed.

Let $\{g_l(C)\}_{l=1}^{L}$ represent the set of unobserved variables (potentially empty) underlying time trend $T$, in which $C$ is assumed to follow a uniform distribution over the considered period. The data points associated with $C$ are assumed to be evenly sampled at a specific frequency, making $C$ the time index. Furthermore, we define that for each variable $V_i$, its parents are denoted by $\text{PA}^i$, and the local causal processes are represented by the SEM below:

$$V_i = f_i \left( \text{PA}^i, \mathbf{g}^i(C), \theta_i(C), \varepsilon_i \right) \tag{1}$$

Here, $\mathbf{g}^i(C) \subseteq \{g_l(C)\}_{l=1}^{L}$ signifies the unobserved variables influencing $T_i$ (empty when no intrinsic trend is present behind $V_i$), while $\theta_i(C)$ represents the effective parameters within the model, also presumed to be functions of $C$. $\varepsilon_i$ denotes a disturbance term, independent of $C$ and exhibiting non-zero variance (i.e., the model is non-deterministic). The mutual independence of $\varepsilon_i$ is also assumed. Note that, the above function (1) is for variables without trends or affected by intrinsic trends only. For those influenced by measurement trends, the real variable and observed variable can be represented by the function (2) and function (3) below, respectively:

$$\underline{V_i} = f_i \left( \text{PA}^i, \varepsilon_i \right) \tag{2}$$

$$V_i = f_i \left( \underline{V_i}, \mathbf{g}^i(C), \theta_i(C), \varepsilon_i \right) \tag{3}$$

In this work, we consider $C$ as a random variable, yielding a joint distribution over $\mathbf{V} \cup \{g_l(C)\}_{l=1}^{L} \cup \{\theta_m(C)\}_{m=1}^{n}$. We assume that this distribution adheres to the Markov and faithfulness properties with respect to the graph resulting from the subsequent modifications to $G$ ($G$ represents the causal structure over $\mathbf{V}$): add $\{g_l(C)\}_{l=1}^{L} \cup \{\theta_m(C)\}_{m=1}^{n}$ to $G$, and for each $i$, add an arrow from each variable in $\mathbf{g}^i(C)$ to $V_i$ and add an arrow from $\theta_i(C)$ to $V_i$. This extended graph is denoted as $G^{aug}$. Evidently, $G$ is merely the induced subgraph of $G^{aug}$ over $\mathbf{V}$.

**Assumption 2** (Causal Markov condition and faithfulness)**.** The joint distribution over $\mathbf{V} \cup \{g_l(C)\}_{l=1}^{L} \cup \{\theta_m(C)\}_{m=1}^{n}$ is Markov and faithful to the augmented graph $G^{aug}$.

To enhance clarity and comprehensibility, this work concentrates on instantaneous causal relationships, and the strength of the causal relations does not change over time. Nevertheless, it is worth noting that our framework can be naturally generalized to encompass time-delayed causal relations, akin to how constraint-based causal discovery has been adapted to manage this (see, e.g. (Chu et al., 2008)).

We further assume that variables influenced by trends do not function as leaf nodes, where leaf nodes are defined as having no descendants. As depicted in Figure 3, a critical difference exists

between intrinsic and measurement trends in their interactions with the underlying causal network; intrinsic trends can be incorporated into this network, whereas measurement trends cannot. This distinction is crucial for our algorithm's capability to differentiate between these trend types. However, when a trend-influenced variable is a leaf node, its trend, whether intrinsic or measurement, is unable to integrate into the existing causal network. Therefore, distinguishing between intrinsic and measurement trends becomes problematic in such cases, as both types exhibit similar characteristics.

# 4 THE PROPOSED ALGORITHM

In this section, we introduce the proposed algorithm, TrendDiff, designed to identify all variables influenced by trends (phase 1) and distinguish between those affected by measurement trends and those exhibiting intrinsic trends (phase 2).

## 4.1 PHASE 1: DETECTION OF TREND-INFLUENCED VARIABLES AND CAUSAL STRUCTURE RECOVERY

In this section, we leverage the detection of changing causal modules to detect variables exhibiting time trends and deduce the causal network for $\mathbf{V} \cup \{C\}$. The core concept hinges on using the (observed) variable $C$ as a surrogate for the unobserved $\{g_l(C)\}_{l=1}^L \cup \{\theta_m(C)\}_{m=1}^n$. In essence, we utilize $C$ to encapsulate the $C$-specific information. Under the assumptions in Section 3, it is feasible to deploy conditional independence tests on the combined set of $\mathbf{V} \cup \{C\}$ to detect variables with time trends and recover the structure. This is achieved by Algorithm 1 and supported by Theorem 1.

In Algorithm 1, we first construct a complete undirected graph, denoted $U_C$, which incorporates both $C$ and $\mathbf{V}$. In Step 2 of the algorithm, the decision regarding whether a variable $V_i$ exhibits a time trend is contingent upon the conditional independence between $V_i$ and $C$, given a subset of other variables. If a time trend is present in $V_i$, then the module of $V_i$ evolves in conjunction with $C$. Consequently, the probability distribution of $V_i$ given its non-$C$ parents, namely $P\left(V_i \mid \mathrm{PA}^i \setminus \{C\}\right)$, will not remain constant across different values of $C$. As a result, $V_i$ and $C$ are conditionally dependent regardless of any subset of variables. Based on this, we assume that if $V_i \perp\!\!\!\perp C \mid \mathrm{PA}^i \setminus \{C\}$, there should be no time trend in $V_i$. Conversely, if this assumption does not hold, then we claim to detect variables with time trends. After this, all variables linked to $C$, referred to as "$C$-specific variables", are considered to be with time trends. Step 3 aims to discover the skeleton of the causal structure over V. It leverages the results from Step 2: if neither $V_i$ nor $V_j$ is adjacent to $C$, then $C$ does not need to be involved in the conditioning set. In practice, one may apply any constraint-based search procedures on $\mathbf{V} \cup C$, e.g., SGS (the Spirtes, Glymour and Scheines algorithm) and PC (the Peter-Clark algorithm) (Spirtes et al., 1993). Its (asymptotic) correctness is justified by the following theorem 1. Finally, step 4 is employed to orient the obtained skeleton based on both standard orientation rules and distribution shift. For a comprehensive explanation of the step 4 orientation procedure and the complete proof of Theorem 1, please refer to (Huang et al., 2020).

---

**Algorithm 1** Detection of Time-trend Variables and Recovery of Causal Structure

---

1: Build a complete undirected graph $U_{\mathcal{G}}$ on the variable set $\mathbf{V} \cup C$.
2: (Detection of time-trend variables) For each $i$, test for the marginal and conditional independence between $V_i$ and $C$. If they are independent given a subset of $\{V_k \mid k \neq i\}$, remove the edge between $V_i$ and $C$ in $U_{\mathcal{G}}$.
3: (Recovery of causal skeleton) For every $i \neq j$, test for the marginal and conditional independence between $V_i$ and $V_j$. If they are independent given a subset of $\{V_k \mid k \neq i, k \neq j\} \cup \{C\}$, remove the edge between $V_i$ and $V_j$ in $U_{\mathcal{G}}$.
4: (Orientation) For the obtained skeleton, orient it by standard orientation rules and distribution shift. After the orientation process, we can get the causal network for $\mathbf{V} \cup C$, called $G^{\mathrm{phase1}}$.

---

**Theorem 1:** *Given Assumptions made in Section 3, for every $V_i, V_j \in \mathbf{V}$, $V_i$ and $V_j$ are not adjacent in G if and only if they are independent conditional on some subset of $\{V_k \mid k \neq i, k \neq j\} \cup \{C\}$.*

## 4.2 PHASE 2: UTILIZING STRUCTURAL DIFFERENCES TO DISTINGUISH BETWEEN INTRINSIC AND MEASUREMENT-TREND VARIABLES

After Phase 1, we procured the set of variables exhibiting time trends (those associated with $C$) as well as the causal network $G^{phase1}$ for $\mathbf{V} \cup C$. In phase 2, we demonstrate that by examining the different structures within causal networks, it is feasible to differentiate variables with intrinsic trends from those influenced by measurement trends.

### 4.2.1 DISTINGUISH BETWEEN INTRINSIC AND MEASUREMENT TRENDS BY $G^{phase1}$

As depicted earlier, intrinsic-trend variables do not change the causal network, whereas those variables characterized by measurement trends can induce structural alterations in causal discovery from the observed variables. Next, we delve into how a measurement-trend variable influences the causal structure of $G^{\text{phase1}}$ and leverage this understanding to partly distinguish between the two trend types.

**Figure 4** illustrates how a measurement-trend variable alters the output causal structure of Phase 1. In **Figure 4 (a)**, we depict a chain with a measurement trend in $X_2$. During Phase 1, the time index $C$ is integrated into our analysis to pinpoint all trend variables. Due to the presence of a measurement trend in $X_2$, a connection from $C$ to $X_2$ is established. Furthermore, based on the conditional independence observed in the actual structure Figure 4(a), we have $T \perp\!\!\!\perp X_3$ and, crucially, $T \not\perp\!\!\!\perp X_3 | X_2$. By extension, because $C$ is a proxy for $T$, the relationships $C \perp\!\!\!\perp X_3$ and $C \not\perp\!\!\!\perp X_3 | X_2$ should hold. Therefore, $X_2$ is a collider and the direction is from $X_3$ to $X_2$. The dependency dynamics between $X_1$ and $C$ follow suit. As a result, the Phase 1 structural outcome for observed variables should be the one shown in **Figure 4 (b)**. It's worth noting that since the measurement trend $T$ is independent across all variables within the real causal network, no arrow can stem from the measurement-trend variable to other variables in $G^{\text{phase1}}$ (cause the observed measurement-trend variable $X_2$ would always be identified as a collider). In essence, any linkage from a "C-specific variable" to other entities indicates an intrinsic trend.

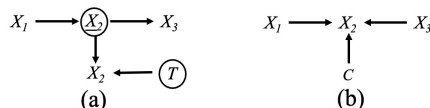

(a)  (b)

Figure 4: An illustration of how a measurement-trend variable alters the output causal structure of Phase 1. (a) the real structure with a measurement trend in $X_2$. (b) the output structure.

In summary, we first employ $G^{\text{phase1}}$ to discern intrinsic-trend variables. A "C-specific variable" is deemed to exhibit an intrinsic trend if it possesses any arrow pointing to other variables in $G^{\text{phase1}}$.

### 4.2.2 DISTINGUISH BETWEEN INTRINSIC AND MEASUREMENT TRENDS BY FURTHER TESTS

Having identified certain intrinsic-trend variables based solely on the structure of $G^{\text{phase1}}$, it becomes necessary to undertake additional conditional independence tests for further recognition of other intrinsic-trend variables. As illustrated in Figure 3, the children of time-trend variables serve as critical pivot points in their differentiation process. For variables with intrinsic trends (see Figure 3b), there is $T_2 \not\perp\!\!\!\perp X_3$ and $T_2 \perp\!\!\!\perp X_3 | X_2$. Conversely, for variables with measurement trends (see Figure 3c), there is $T_2 \perp\!\!\!\perp X_3$ and $T_2 \not\perp\!\!\!\perp X_3 | X_2$. Thus, the criterion for identifying an intrinsic-trend variable $X_2$ can be $T_2 \not\perp\!\!\!\perp X_3$ and $T_2 \perp\!\!\!\perp X_3 | X_2$. Here $T_2$ is the trend of $X_2$ and $X_3$ is a child of $X_2$. Since the trend $T_2$ is not directly observable in this context. As an alternative, we employ the time index $C$ again, working as a suitable proxy for the unobservable trend. Therefore, the criterion is: $C \not\perp\!\!\!\perp X_3$ and $C \perp\!\!\!\perp X_3 | X_2$.

The first row of **Figure 5** illustrates four scenarios of child variables that may arise when screening for the intrinsic-trend variable $X_1$. In **Figure 5 (a)**, no trend is evident in the child variable $X_2$, allowing us to easily identify $X_1$ as an intrinsic-trend variable using our criterion. However, **Figure 5 (b)(c)**, the child variable $X_2$ exhibits intrinsic and measurement trends, respectively. Since trends are functions of time, time serves as a confounder (common cause) of trends $T_1$ and $T_2$. In these cases, the path from $T_1$ to $X_2$ via the confounder "time" cannot be blocked, as neither "time" nor $T_2$ is observable (we can obtain a surrogate for $T_2$, but it is insufficiently accurate to block the path).

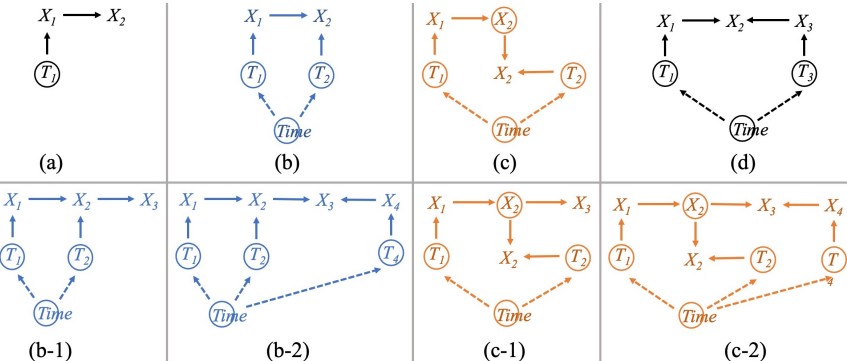

Figure 5: Different scenarios for descendants of intrinsic-trend variables. First row: Four possible cases of intrinsic-trend variable's child nodes in causal networks. (a) Child node without trend. (b) Child node with an intrinsic trend. (c) Child node with a measurement trend. (d) Child node with a trend from other observable nodes. Second row (b-1), (b-2), (c-1), and (c-2): Four possible cases of intrinsic-trend variable's second-order descendant for structure (b) and (c).

Consequently, we cannot distinguish variables with intrinsic trends from those with measurement trends when all child variables have trends. However, if the trend in the child variable $X_2$ originates from its other observable parent $X_3$, as depicted in **Figure 5 (d)**, the intrinsic-trend variable $X_1$ is identifiable since we can block the path through "time" by conditioning on $X_3$.

For structures (b) and (c), first-order descendants (children) do not facilitate distinguishing trend types. However, can second-order descendants provide clarity? Will it help if structures similar to (a) or (d) emerge subsequent to (b) and (c)? The subsequent row illustrates potential second-order descendant structures for both (b) and (c). Although **Figure 5 (b-1)(b-2)** remain non-identifiable, **Figure 5 (c-1)(c-2)** can be discerned. The principles behind (c-1) and (c-2) align with those of (a) and (d), namely $C \not\perp\!\!\!\perp X_3$ and $C \perp\!\!\!\perp X_3 | X_1$. It's noteworthy that structures (c-1) and (c-2) essentially represent (a) and (d) but with an added measurement-trend variable subsequent to the intrinsic-trend variable $X_1$ under examination. Extending this, we can infer that all structures obtained by adding $n$ measurement-trend variables between $X_1$ and $X_2$ in structures (a) and (d) can theoretically be identified, where $n=0,1,2...$

In summary, intrinsic-trend variables are discernible in this process only when (1) the intrinsic-trend variable $X$ to be tested possesses at least one descendant variable $Y$ without trends (like structure (a)) or with trends stemming from other observable variables (like structure (d)); and (2) there are no other intrinsic-trend variables on the path from $X$ to $Y$. Algorithm 2 for Phase 2 is provided in Appendix A.2. By combining Algorithm 1 and 2, we can obtain the proposed TrendDiff algorithm.

## 5 EXPERIMENTS

### 5.1 SIMULATIONS

**Fixed structure.** Synthetic datasets were constructed based on the SEMs described in Appendix Figure 8. Variables $X_1$, $X_2$, and $X_7$ were specifically designed to show intrinsic trends, while $X_3$ and $X_6$ exhibited measurement trends. All trends were modeled as sinusoidal functions with periods $w$ chosen randomly from a uniform distribution $\text{Unif}([5, 25])$. The relationships in the dataset were set to be nonlinear, with half of the links following $f^{(1)}(x) = \left(1 - 4e^{-x^2/2}\right) x$ and the remaining half following $f^{(2)}(x) = \left(1 - 4x^3 e^{-x^2/2}\right) x$. Noise types (Gaussian, Exponential, Gumbel) and various sample sizes (T = 600, 900, 1200, 1500) were incorporated into the simulations. After data generation, we employed TrendDiff to identify variables with intrinsic trends. Additionally, we compared the performance of the PC algorithm on datasets before and after the removal of measurement trends identified by TrendDiff. The effectiveness was quantitatively assessed using F1 score, precision, and recall metrics. Each experimental configuration was repeated across 50 trials to ensure robustness.

The results from the fixed structure simulations are illustrated in **Figure 6**. **Figure 6(a)** showcases the efficacy of TrendDiff in detecting variables influenced by intrinsic trends. An increase in the length of the dataset correlates with improvements in the algorithm's performance. Specifically, for datasets of 1500 data points or more, the algorithm achieves near-optimal efficiency, with all primary metrics nearing 0.9. Additionally, TrendDiff maintains consistent performance across various types of noise, demonstrating its robustness. **Figure 6(b)** provides a comparative analysis of the PC algorithm's performance on datasets both before and after the removal of identified measurement trends. The removal of these trends markedly improves the performance of the PC algorithm.

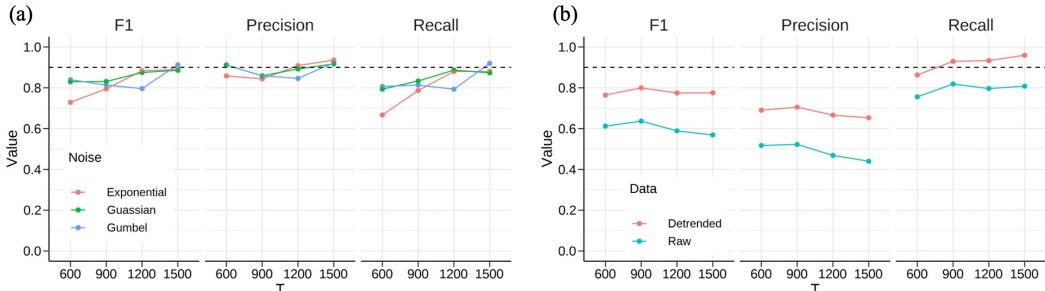

Figure 6: Simulation performance. (a) Performance of identifying intrinsic-trend variables. (b) Performance of PC algorithm using data pre and post-elimination of detected measurement trends.

**Random structure.** In addition to the fixed-structure simulations, the TrendDiff algorithm was also assessed using datasets generated from random structures. This random structure simulation maintained the same parameter settings as its fixed structure counterpart, with the exception that the underlying causal network was randomly generated rather than predetermined. The outcomes of the random structure simulations are detailed in the appendix. Specifically, Figure 11 illustrates the algorithm's performance in identifying intrinsic-trend variables across a range of data lengths (T) and noise types. In line with the findings from the fixed structure simulations, TrendDiff demonstrates robustness against various types of noise. Additionally, a consistent trend is observed where the performance of the algorithm improves as the data length increases. Further insights into the stability of our method are illustrated in Figure 12, which demonstrates TrendDiff's resilience in various data dimensions and sparsity levels. Figure 13 evaluates TrendDiff's performance in scenarios involving linear trends. The results show that TrendDiff is particularly proficient in linear-trend scenarios, further highlighting its effectiveness in a broad range of conditions. When tackling practical issues, considering computational complexity becomes essential. The computational efficiency of TrendDiff is demonstrated in Figure 14, which displays the processing times across different data sizes and number of nodes. Notably, the analysis revealed a non-linear increase in processing times with the growth in data length. However, it is important to point out that, even with this escalation for larger datasets, the processing duration stays within a feasible range for practical applications.

## 5.2 REAL DATA

We also applied the proposed approach to a real environmental health dataset. This dataset contains daily values of variables regarding air pollution, weather, and sepsis emergency hospital admission in Hong Kong for the period from 2007 to 2018. It is a typical dataset used to assess the interactions between environmental factors and human health. There are pronounced time trends in this data (**Figure 7a**), rendering it a good application example for the TrendDiff algorithm. In our initial analysis, we applied TrendDiff to determine the intrinsic trend variables within the data. The outcome from Phase 1 indicates that sepsis emergency hospital admissions, $CO$, $O_3$, and $NO_2$ are variables exhibiting a trend, be it measurement or intrinsic. Subsequently, in phase 2, we differentiated between measurement-trend and intrinsic-trend variables. It was discerned that $CO$, $O_3$, and $NO_2$ have intrinsic trends while the daily count of sepsis emergency hospital admissions stood out as the sole variable characterized by a measurement trend. This result is consistent with existing evidence. There have been heated discussions in top medical journals about the observed rise in sepsis cases. A prevailing consensus among researchers is that this uptick in sepsis incidences can be largely attributed to the refined definitions and enhanced coding practices for sepsis, rather than the real incidence increase (Rhee et al., 2017; Fleischmann-Struzek et al., 2018). As for the trio of variables

recognized with an intrinsic trend — $CO$, $O_3$, and $NO_2$ — ample research has been conducted on their time trends. However, none have ascribed these trends to measurement inaccuracies, supporting our results here (Wei et al., 2022).

Beyond simply distinguishing between intrinsic and measurement trends, we also compared causal discovery outcomes before and after the removal of the identified measurement trends. Here we utilized the Peter-Clark-momentary-conditional-independence plus (PCMCI+) method, a well-regarded causal discovery algorithm for time series (Runge, 2020). This dataset under scrutiny was a typical environmental health dataset from Hong Kong, with a focus on uncovering environmental factors contributing to sepsis. As depicted in **Figure 7(b)**, the impact of eliminating the identified measurement trend is notably significant on the causal analysis results. Our initial analysis, based on the raw data, classified $CO$ and $SO_2$ as potential mitigating factors against sepsis. However, when the measurement trend associated with sepsis was removed, the analysis showed a different picture. It revealed that temperature alone was a risk factor for sepsis, which is supported by existing evidence (Helbing et al., 2022). Though this analysis did not deal with other factors like seasonality, the notable differences in the findings underscore the critical importance of detecting and correcting measurement trends in causal analysis.

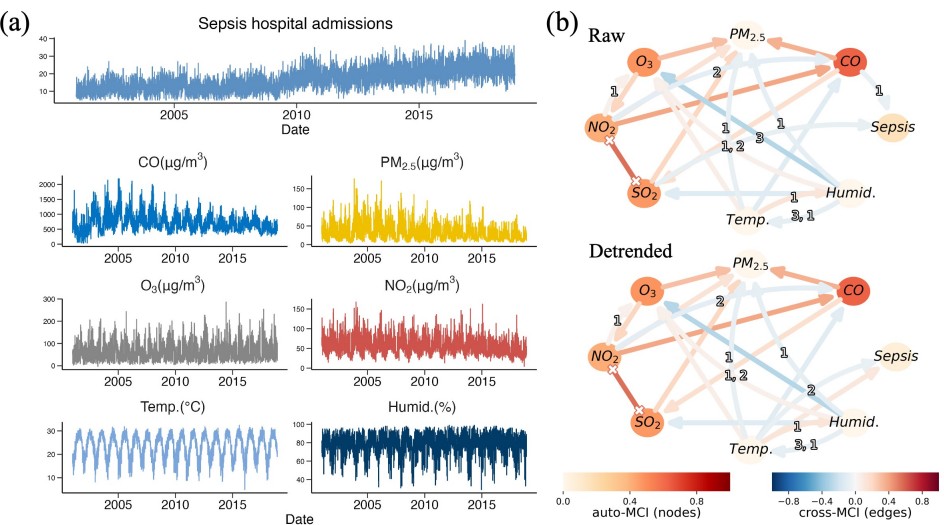

Figure 7: Evaluation of performance using a real-world dataset. (a) Depiction of time series variables. (b) Raw: discovery of structure from raw data by Peter-Clark-momentary-conditional-independence plus (PCMCI+). Detrended: discovery of structure after removal of identified measurement trends by PCMCI+. Here a curved arrow represents a lagged causal relationship, with the lag day shown on the curve. A straight arrow means a contemporaneous association. A straight line terminating in crosses at both ends represents contemporaneous adjacency with unresolved directionality stemming from contradictory orientation rules. The link color refers to the cross-MCI value, which indicates the strength of the relationships. The node color denotes the auto-MCI value, representing how strong the autocorrelation is.

## 6 CONCLUSION AND DISCUSSIONS

The need to discern intrinsic trends from measurement trends has been a longstanding challenge. TrendDiff, our innovative algorithm, is tailored to address this difficulty as evidenced by its successful application in both simulated and real-world scenarios. However, we recognize a few limitations of this algorithm. Firstly, although we assume trend-influenced variables are non-leaf nodes, differentiating trend types in leaf nodes also holds value. Secondly, in reality, intrinsic and measurement trends may coexist in variables, a scenario that TrendDiff currently cannot handle. We leave improving TrendDiff's ability to differentiate trends in leaf nodes and mixed types for future work.

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

# A  APPENDIX / SUPPLEMENTAL MATERIAL

## A.1  ASSUMPTIONS

### A.1.1  UNDERSTANDING PSEUDO CAUSAL SUFFICIENCY THROUGH A SIMPLIFIED EXAMPLE

In environmental epidemiology, time series analyses are frequently employed to evaluate the immediate effects of air pollution on respiratory health. Here, the exposure of interest is the daily concentration of air pollutants, while the outcome is the daily number of hospital admissions for respiratory diseases. The primary objective is to determine whether day-to-day fluctuations in air pollutant levels influence the day-to-day variation in respiratory-related hospital admissions. Typically, factors like temperature and relative humidity are adjusted for as known confounders in these studies.

Nonetheless, in addition to these controlled variables, there exist other unobserved confounders that can significantly impact the analysis. These include variables such as socioeconomic status, changes in policy, and seasonal variations. Given that the duration of this kind of time series studies often spans several years or more, it's evident that these unobserved confounders could have a substantial effect on the findings. Therefore, it's crucial to acknowledge and attempt to account for these unseen factors to ensure the robustness and accuracy of the results.

In this scenario, the notion of pseudo causal sufficiency emerges. Traditionally, it's assumed in such studies that any unobserved confounders are associated with time and can be encapsulated as a smooth function of time, typically represented through splines. By incorporating this spline function into the analysis, the model effectively accounts for unobserved confounders. This assumption has been widely adopted across environmental health studies. However, it's important to note that this example is provided for the sake of understanding pseudo causal sufficiency more easily. In reality, these traditional studies do not explicitly introduce this concept.

### A.1.2  NON-TIME-DELAYED RELATIONSHIPS

In real-world scenarios, both non-time-delayed and time-delayed causal relationships are crucial. In theory, all causal relationships involve some form of delay (as we often say, cause precedes effect). However, in real life, due to our limited knowledge of the true nature of various relationships and the limited precision of the data available to us, we often observe relationships that appear to have no delay. For example, the true mechanism by which air pollution affects lung function may involve a delay of one hour—meaning exposure to severe air pollution now might lead to weakened lung function an hour later. However, since we only have access to daily data on air pollution and lung function, this inherently delayed relationship may appear in the data as if there is no delay. Thus, in real life, both non-time-delayed and time-delayed relationships are very common.

In our study, the assumption of no time-delayed relationships is partly made for the sake of clarity and readability of the paper. As this is the initial introduction of intrinsic and measurement trends using graphical models, and the first formal presentation of the differentiation challenge, clarity is paramount. Some of the figures (Figure 5 for example) in the manuscript are already quite complex and difficult to understand without considering delays. If time delays were to be taken into account, the number of nodes would increase manifold, severely affecting the presentation of the problem and the algorithm.

### A.1.3  TREND-INFLUENCED VARIABLES ARE NON-LEAF NODES

The impact of measurement trends on causal discovery outcomes is much smaller for leaf nodes compared to non-leaf nodes. This is why we believe that our method remains highly valuable even if it cannot differentiate the trend types in leaf nodes. As described in paragraph 3 and Figure 2, measurement trends introduce two issues for constrained-based causal discovery algorithms: 1. the dependence between measurement-trend variables and their neighbors weakens with increasing trends; 2. the conditional independence given the measurement-trend variable vanishes, yielding increasing dependence. For non-leaf nodes, such as $X_2$ and $X_3$ in Figure 2, both these two issues are present. In contrast, leaf nodes like $X_1$ and $X_4$ in the same figure are only subject to the first issue. This is because a leaf node, having no children, does not act as a mediator in any relationship.

## A.2 ALGORITHM 2

---

**Algorithm 2** Identify intrinsic-trend variables by structural differences

---

**Require:** Dataset $\mathbf{V}$, "$C$-specific variables" identified in phase 1, causal structure $G^{\text{phase1}}$ identified in phase 1, significance threshold $\alpha$, conditional independence test $\text{CI}(X, Y, \mathbf{Z})$ returning $p$-value.
**Ensure:** The set of variables exhibiting an intrinsic trend and the set of variables demonstrating a measurement trend within $\mathbf{V}$.
1: IntrinsicSet = $\emptyset$
2: **for all** $X_i \in$ "$C$-specific variables" **do**
3:     $\boldsymbol{\beta}$ = Causal Graph Matrix($G^{\text{phase1}}$)
4:     $links = \boldsymbol{\beta}_i.$
5:     **if** $1 \in links$ **then**                   ▷ Check if $X_i$ has outgoing links
6:         Store $X_i$ in IntrinsicSet
7: RestSet = "$C$-specific variables" - IntrinsicSet
8: **for all** $X_i \in$ RestSet **do**
9:     TestNodes = $\mathbf{V}$ - "$C$-specific variables"
10:     **for all** $X_j \in$ TestNodes **do**
11:         JNb = Neighbors($X_j$) - $X_i$
12:         **for all** $n \in$ Range(len(JNb)) **do**
13:             **for all** $S_0 \in$ Combinations(JNb, $n$) **do**
14:                 $S_1 = S_0 + X_i$
15:                 $C$ = Time index
16:                 $p_0 = \text{CI}(C, X_j, \mathbf{S_0})$
17:                 $p_1 = \text{CI}(C, X_j, \mathbf{S_1})$
18:                 **if** $(p_0 < \alpha)$ & $(p_1 > \alpha)$ **then**
19:                     Store $X_i$ in IntrinsicSet
20: MeasurementSet = "$C$-specific variables" - IntrinsicSet
21: **return** IntrinsicSet, MeasurementSet

---

**Note:** the Causal Graph Matrix in line 3 outputs a Causal Graph object, where $\boldsymbol{\beta}_{j,i} = 1$ and $\boldsymbol{\beta}_{i,j} = -1$ indicate i $\rightarrow$ j; $\boldsymbol{\beta}_{i,j} = \boldsymbol{\beta}_{j,i} = -1$ indicates i — j; $\boldsymbol{\beta}_{i,j} = \boldsymbol{\beta}_{j,i} = 1$ indicates i $\leftrightarrow$ j.

## A.3 FIXED STRUCTURE SIMULATION

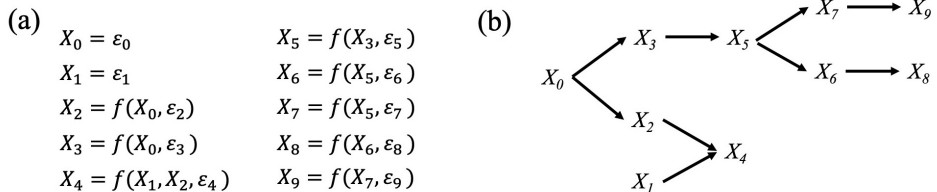

(a)
$$X_0 = \varepsilon_0 \qquad X_5 = f(X_3, \varepsilon_5)$$
$$X_1 = \varepsilon_1 \qquad X_6 = f(X_5, \varepsilon_6)$$
$$X_2 = f(X_0, \varepsilon_2) \qquad X_7 = f(X_5, \varepsilon_7)$$
$$X_3 = f(X_0, \varepsilon_3) \qquad X_8 = f(X_6, \varepsilon_8)$$
$$X_4 = f(X_1, X_2, \varepsilon_4) \qquad X_9 = f(X_7, \varepsilon_9)$$

Figure 8: Data structure for fixed-structure simulation. (a) The SEMs according to which we added intrinsic and measurement trends and generated the simulated data. (b) The visualization of the structure. All relationships are nonlinear.

The process to generate simulation data for assessing the TrendDiff algorithm in a fixed structure context involves three primary steps:

1. Original Structure Acquisition: The baseline fixed structure, void of any trends, is depicted in **Figure 8**. This figure is subdivided into two parts: Figure 8(a), which represents the Structural Equation Models (SEMs), and Figure 8(b), which illustrates the structure's visualization.

2. Trend Integration: To incorporate both identifiable structures from Figure 5(a) and (d) into the simulation, intrinsic trends were embedded in variables $X_1$, $X_2$, and $X_7$. Additionally, to emulate real-world data characteristics, measurement trends were introduced in $X_3$ and $X_6$. These trends were modeled as smooth functions of time, formulated as $trend = \sin\left(\frac{w \cdot t}{T}\right)$, where the period $w$ is

randomly drawn from a uniform distribution $\mathrm{Unif}([5, 25])$, $T$ represents the data length, and $t$ is the time index.

3. Data Generation and Testing: The final step involved generating simulation data based on the modified structure from the above two steps. All relationships in the data are set to be nonlinear, with 50% of the links using the function $f^{(1)}(x) = \left(1 - 4e^{-x^2/2}\right) x$ and the other 50% employing $f^{(2)}(x) = \left(1 - 4x^3 e^{-x^2/2}\right) x$. The algorithm's performance was evaluated under a variety of noise distributions (Gaussian, Exponential, Gumbel) and different sample sizes (T = 600, 900, 1200, 1500). For each scenario, the TrendDiff algorithm was tested using the generated data, with 50 trials conducted in each setting to ensure statistical robustness.

The efficacy of the TrendDiff algorithm is quantitatively assessed using three key metrics: F1 score, precision, and recall. These metrics are defined as follows:

Precision: This metric calculates the proportion of true positive outcomes among the total predicted positives. Mathematically, it is expressed as the ratio of the number of true positives (TP) to the sum of true positives and false positives (FP), given by the formula:

$$P = \frac{TP}{TP + FP}$$

Recall: Also known as sensitivity, this metric measures the proportion of actual positives correctly identified. It is calculated as the ratio of true positives to the sum of true positives and false negatives (FN), described by:

$$R = \frac{TP}{TP + FN}$$

F1 Score: The F1 score is the harmonic mean of precision and recall, providing a single metric that balances both. It is particularly useful in situations where there is an uneven class distribution. The F1 score is computed using the formula:

$$F1 = 2 \times \frac{P \times R}{P + R}$$

These metrics offer a comprehensive evaluation of TrendDiff's performance, effectively capturing its accuracy and robustness in various testing scenarios.

Our study extends to evaluating the performance of the PC (Peter-Clark) algorithm both before and after the removal of measurement trends identified by TrendDiff. This comparative analysis highlights the effectiveness of our methodology in enhancing causal discovery through data preprocessing. In our approach to handle variables affected by measurement trends, we utilized the Savitzky-Golay filter. This process involves subtracting the trend, as determined by the Savitzky-Golay filter, from the original data to produce detrended data. The Savitzky-Golay filter is a well-known polynomial smoothing technique that fits a polynomial of a specified degree to consecutive data points within a moving window, using linear least squares regression. Once the polynomial is fitted, the filter can provide either a smoothed estimate or the derivative of the fitted function. This filtering method is prevalent in various domains such as analytical chemistry and signal processing, especially for dealing with noisy data. The two primary parameters governing the Savitzky-Golay filter are the window size and the polynomial order. The window size determines the number of data points used for each polynomial fit, thereby influencing the degree of smoothing. On the other hand, the polynomial order specifies the complexity of the model used in the fitting process.

Central to our algorithm is the utilization of a nonparametric conditional independence test, which is crucial given the often unknown and highly nonlinear nature of time trends. In this study, we adopted the Kernel-based Conditional Independence test (KCI-test) (Zhang et al., 2012), a method adept at capturing complex nonlinear dependencies. A critical aspect of the KCI-test is the kernel width parameter $w$, integral to constructing kernel matrices and subsequently influencing the performance of the test. We conducted extensive evaluations to determine the optimal kernel width, varying data lengths $T$ and kernel widths $w$. These variations in performance, based on different values of $T$ and $w$, are elucidated in **Figure 9**. The data reveals that as $T$ increases, so does performance efficiency. Notably, a kernel width of $w = 0.5$ consistently yields impressive results, regardless of the $T$ value.

Our findings are in concordance with the recommendations from the original KCI paper, which suggest specific kernel width settings based on sample sizes: set $w$ to 0.8 for sample sizes $n \leq 200$, to 0.3 if $n > 1200$, and to 0.5 in all other instances. In alignment with these guidelines, our study adopted these kernel width configurations, optimizing our approach for varying data scenarios. This methodology underscores our commitment to leveraging advanced statistical techniques for accurate and efficient data analysis. The results from the fixed structure simulations with 90% confidence interval are illustrated in **Figure 10**.

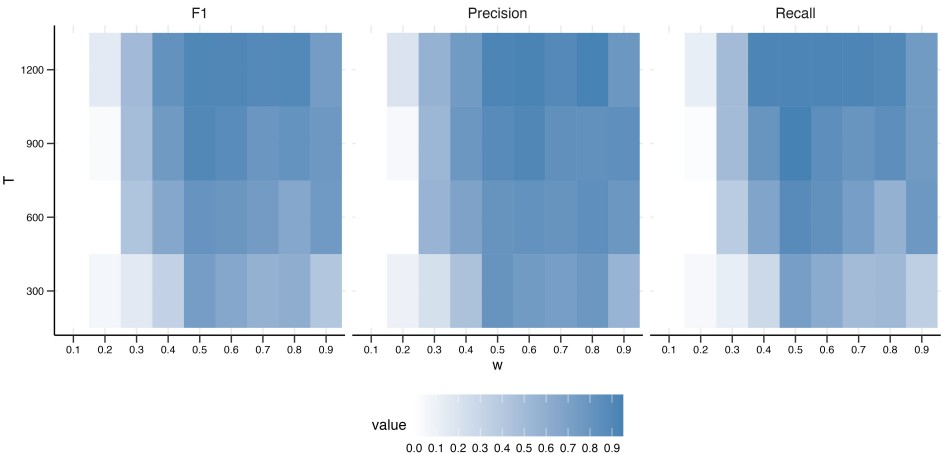

Figure 9: Parameter choosing results. Performance of our algorithm under different kernel width $w$ with changing data length $T$.

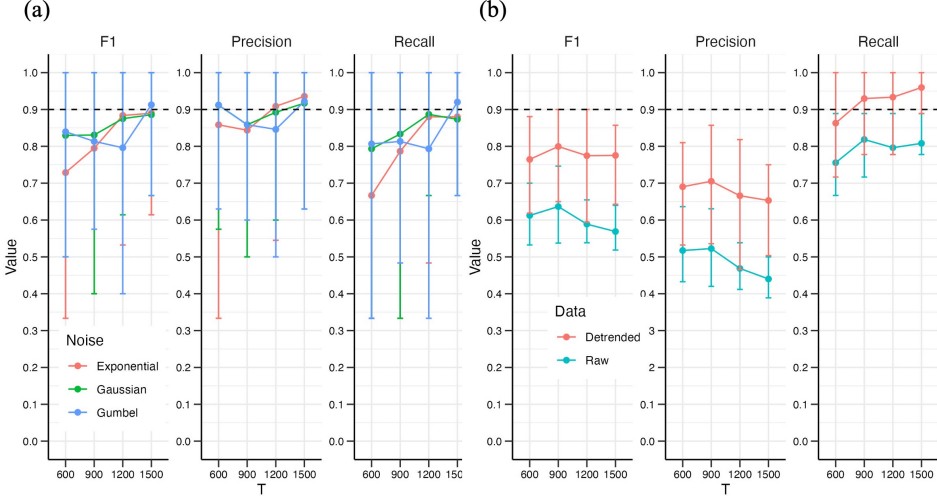

Figure 10: Simulation performance with 90% confidence interval. (a) Performance of identifying intrinsic-trend variables. (b) Performance of PC algorithm using data pre and post-elimination of detected measurement trends.

### A.4 RANDOM STRUCTURE SIMULATION

We also tested our algorithm using simulated data based on random structures. There are three steps to this process: 1) We generated random graph G from the Erdös-Rényi (ER) random graph model, with edges added independently with equal probability. The degree, that is, the total number of edges linked with each node (in + out), is $d$. Given G, the weights of edges are drawn from

$\text{Unif}([-0.6, -0.2] \cup [0.2, 0.6])$ to obtain a weight matrix $W_0$. 2) Given $W_0$, intrinsic and measurement trends are randomly assigned to variables, with $W_0$ updated to $W$. Note that, only intrinsic trend structures like (a) and d in Figure 5 will be generated in this process, which means: a) no trend in leaf nodes; b) variables with trends are not adjacent. 3) Then we sampled $X = W^T X + z \in \mathbb{R}^d$ from noise model. Finally, we generated random datasets $\mathbf{X} \in \mathbb{R}^{n \times d}$ by generating T rows I.I.D. We considered different model setups for noise types, data length T, data dimension, and the degree of sparsity to comprehensively test our algorithm. For each scenario, all metrics precision, recall, and F1 score are computed across all graphs from 50 realizations of the random graph-generating model at data length T in (600, 900, 1200, 1500).

**Figure 11** showcases the performance metrics – F1 score, precision, and recall – for identifying intrinsic-trend variables across different data lengths $T$ and noise types. Notably, the method proves robust across noise variations and, consistent with fixed structure results, performance improves with increasing data length. **Figure 12** provides further insights into our method's stability, demonstrating its resilience across a range of data dimensions and degrees of sparsity, where dimension is denoted by the number of nodes and sparsity is defined as the degree considering edges in both directions. **Figure 13** shows TrendDiff performance on data generated from random structures with linear trends. We measured the identification of intrinsic-trend variables across different data lengths T and noise types in linear-trend scenarios. TrendDiff excels in scenarios with linear trends. **Figure 14** provides an analysis of the processing times and peak memory required by TrendDiff for handling different data sizes and number of nodes. The TrendDiff algorithm was executed on a high-performance computing (HPC) system, featuring a single 25-core CPU. A significant finding from this deployment is the non-linear increase in processing times corresponding to the augmentation of data length. Although there is a marked escalation in processing duration for larger datasets, it is essential to emphasize that the timeframes remain within a practical and manageable range for real-world applications. Specifically, for a dataset with 10 variables and a data length of T=1500, the processing time is maintained at approximately 1000 seconds (17min). The peak memory requested is stable.

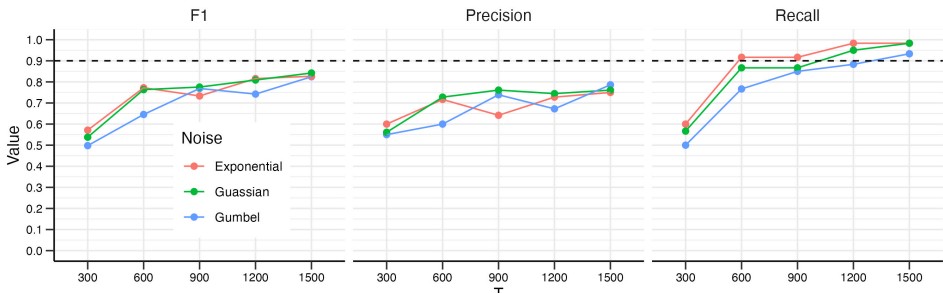

Figure 11: Performance evaluation on data generated from random structures with varying $T$ and noise type. We measure the identification of intrinsic-trend variables across different data lengths $T$ and noise types using F1 score, precision, and recall. Higher values denote better performance.

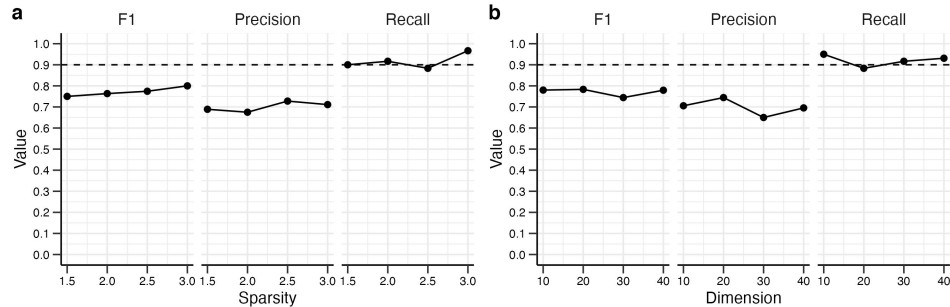

Figure 12: Performance evaluation on data generated from random structures with varying sparsity and dimension. (a) Performance under different sparsity levels. (b) Performance across varying dimensions.

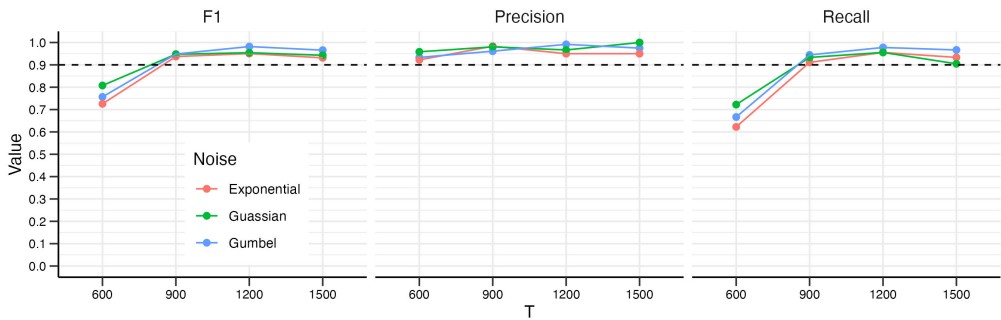

Figure 13: Performance evaluation on data generated from random structures with linear trends. We measure the identification of intrinsic-trend variables across different data lengths $T$ and noise types using F1 score, precision, and recall. Higher values denote better performance.

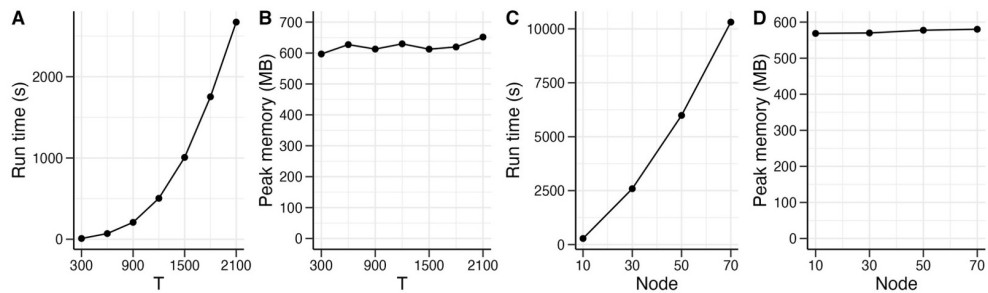

Figure 14: Run time and peak memory requested by TrendDif with increasing data length T and number of nodes.

### A.5 APPLICATION IN REAL-WORLD DATA

Besides simulation studies, we applied our algorithm to a real-world data set about environmental health as well. The data set contains daily values of variables regarding air pollution, weather, and sepsis emergency hospital admission in Hong Kong. This data set is good for exploring the relationships between environmental factors and sepsis. Sepsis, alternatively referred to as septicemia or blood poisoning, is a life-threatening medical emergency when the dysregulated host response to infection injures its own tissues and organs (Singer et al., 2016). It is one of the leading causes of death and contributes significantly to preventable mortality (Organization et al., 2020). In 2017, 11.0 million sepsis-related deaths were reported globally, constituting 20% of all the annual deaths (Rudd et al., 2020). Understanding the relationships between environmental factors and sepsis risk provides a deeper insight into the underlying mechanisms through which environmental factors may predispose, trigger, or exacerbate sepsis conditions. This knowledge is not only pivotal for timely intervention but also offers a foundation for formulating targeted prevention strategies.

Data on daily sepsis emergency hospital admissions of Hong Kong were obtained from the Hospital Authority, which compiles information on all emergency admissions from publicly funded hospitals that provide 24-hour accident and emergency services and cover 90 percent of hospital beds for Hong Kong residents. Sepsis cases were identified based on the ninth version of the International Classification of Diseases (ICD-9: 38), with a total number of 108,831 admissions for a period of 6,543 days spanning from 2007 to 2018.

Hourly concentrations of air pollutants, including carbon monoxide (CO), particulate matter with aerodynamic diameter 2.5m ($PM_2.5$), ozone ($O_3$), and nitrogen dioxide ($NO_2$), were obtained from the general air quality monitoring stations in Hong Kong. For daily $O_3$ concentrations, the maximum 8-hour averages were considered, while 24-hour averages were used for daily concentrations of other air pollutants. The air pollutant data from all monitoring stations were combined to compute city-wide averages for each pollutant. Weather data pertaining to daily average temperature and

relative humidity were acquired from the Hong Kong Observatory. The summary statistics of these variables are shown in Table 1.

Table 1: Summary statistics of daily sepsis emergency hospital admissions, air pollution, and weather in Hong Kong, 2007-2018[a]

| Variables | Mean (SD) | Min | 25th | 50th | 75th | Max |
|---|---|---|---|---|---|---|
| *Outcome (daily count)* | | | | | | |
| Sepsis | 19(6) | 5 | 15 | 19 | 23 | 39 |
| *Air pollution ($\mu g\,m^{-3}$)* | | | | | | |
| CO | 674.9(2322) | 250.0 | 504.0 | 637.3 | 800.6 | 2001.8 |
| PM$_{2.5}$ | 28.9(180) | 4.0 | 14.9 | 24.9 | 38.3 | 138.3 |
| O$_3$ | 61.7(366) | 3.3 | 32.6 | 53.6 | 82.7 | 286.5 |
| NO$_2$ | 52.3(184) | 4.1 | 39.0 | 49.2 | 62.5 | 162.4 |
| *Weather* | | | | | | |
| Temperature (°C) | 23.6(52) | 4.9 | 19.3 | 24.8 | 28.2 | 32.4 |
| Humidity (%) | 78.2(105) | 29.0 | 74.0 | 79.0 | 85.0 | 99.0 |

[a]Abbreviations: SD = standard deviation; min = minimum value; 25th = 25th percentile; 50th = 50th percentile; 75th = 75th percentile; max = maximum value; CO = carbon monoxide; PM$_{2.5}$ = particulate matter with aerodynamic diameter 2.5m; O$_3$ = ozone; NO$_2$ = nitrogen dioxide; Temp. = temperature; Humid. = relative humidity.

In the application of our algorithm to real-world datasets, we began by employing the proposed method to systematically identify the sets of variables exhibiting any trends, then focusing specifically on distinguishing between intrinsic-trend variables and measurement-trend variables. These results were rigorously validated against existing research and literature pertaining to trend behaviors and measurement errors in the context of environmental variables and sepsis data. This analysis served to validate the precision of our algorithm. Following this, we applied the "Peter-Clark-momentary-conditional-independence plus (PCMCI+)" causal discovery algorithm to the datasets, conducting this procedure both prior to and subsequent to the removal of the identified measurement trends. This two-phase application facilitated a comprehensive comparative analysis, effectively highlighting the impact and advantages of our algorithm in enhancing causal discovery processes. The results from this application demonstrate the utility of our algorithm as a potent data preprocessing tool, significantly aiding in the accuracy and efficacy of subsequent causal analysis. The effectiveness of the algorithm in real-world scenarios, especially in complex fields like environmental studies and medical research, emphasizes its versatility and potential for broader applications.

Below we detail the PCMCI+ algorithm:

PCMCI+ belongs to the so-called constraint-based causal discovery methods family, which is based on conditional independence test(Runge, 2020). Here "PC" refers to the developers Peter and Clark, "MCI" means that the momentary conditional independence (MCI) test idea is added to the traditional PC algorithm, and "+" reminds users that it extends the earlier version of PCMCI to include the discovery of contemporaneous links(Runge et al., 2019). Like other causal graphic models, PCMCI+ works under the general assumptions of the causal Markov condition (each variable in the system is independent of its non-descendants, given its parent variables) and faithfulness (probabilistic information in data emerges not by chance but from causal structures) (Runge, 2018). On top of the general assumptions, two specific assumptions are also requested: causal stationarity (i.e., the causal links hold for all the studied time points) and causal sufficiency (i.e. measured variables include all of the common causes).

PCMCI+ algorithm starts with a skeleton discovery phase, which serves to remove the adjacencies due to indirect paths (mediation) and common causes (confounders). This phase can be divided into lagged stage and contemporaneous stage. The former is to identify lagged potential parents, and the latter is to identify contemporaneous potential parents and optimize identified lagged parents. In the lagged stage, for each variable $X_t^j$, a superset of lagged ($\tau > 0$) parents $\widehat{\beta_t^-}\left(X_t^j\right)$ is estimated with the iterative PC1 algorithm. In the contemporaneous stage, we iterate through subsets $\mathcal{S} \subset \boldsymbol{X_t}$ of contemporaneous adjacencies and remove adjacencies for all (lagged and contemporaneous) ordered

pairs $(X_{t-\tau}^i, X_t^j)$ with $X_t^j \in \boldsymbol{X_t}$ and $X_{t-\tau}^i \in \mathbf{X}_t \cup \widehat{\beta_t^-}\left(X_t^j\right)$ if the MCI conditional independence holds: $\left(X_{t-\tau}^i \perp X_t^j \mid \mathcal{S}, \widehat{\beta_t^-}\left(X_t^j\right), \widehat{\beta_{t-\tau}^-}\left(X_{t-\tau}^i\right)\right)$. This skeleton discovery phase returns a skeleton of causal network of undirected relationships among the nodes.

Next in the orientation phase the contemporaneous links (lagged links can automatically be directed by time order) in the recognized skeleton will be oriented by the collider orientation stage and followed by the rule orientation stage. In collider orientation process, unshielded triples $X_{t-\tau}^i \to X_t^k \circ - \circ X_t^j$ (for $\tau > 0$) or $X_t^i \circ - \circ X_t^k \circ - \circ X_t^j$ (for $\tau = 0$) where $X_{t-\tau}^i, X_t^j$ are not adjacent would be oriented as collider structures if $X_t^k$ is not in the sepset $\left(X_{t-\tau}^i, X_t^j\right)$ according to the rule "none". Here sepset $\left(X_{t-\tau}^i, X_t^j\right)$ means the controlled variables when obtaining conditional independence of $X_{t-\tau}^i, X_t^j$. Besides the rule "none", another two rules "conservative" and "majority" can also be chosen in this stage. After that, three rules R1, R2, and R3 are followed to orient left links. R1 rule states that all unambiguous $X_{t-\tau}^i \to X_t^k \circ - \circ X_t^j$ can be oriented as $X_{t-\tau}^i \to X_t^k \to X_t^j$ since there is no collider left in this stage; in R2 rule, all $X_t^i \to X_t^k \to X_t^j$ structures with $X_t^i \circ - \circ X_t^j$ are oriented as $X_t^i \to X_t^j$ to avoid circles. Finally, in R3 rule, for all unambiguous $X_t^i \circ - \circ X_t^k \to X_t^j$ and $X_t^i \circ - \circ X_t^l \to X_t^j$ where $X_t^k, X_t^l$ are independent and $X_t^i \circ - \circ X_t^j$, we orient $X_t^i, X_t^j$ as $X_t^i \to X_t^j$ to satisfy both the no-collider and no-circle rules. After the orientation process, we leave unoriented correlations as $\circ - \circ$ and conflicting correlations as $\times - \times$.

For PCMCI+ analysis, the Python module "tigramite" (version 5.1.0.3) was used. The main free parameters of PCMCI+ (in addition to the free parameters of the conditional independence tests) are the maximum time delay $\tau_{\max}$ and the significance threshold $\alpha_{\mathrm{PC}}$. We used 3 and 0.05 for these two parameters, respectively. In the output causal network produced by PCMCI+, a curved arrow represents a lagged causal relationship, with the lag day shown on the curve. A straight arrow means a contemporaneous association. A conflicting, contemporaneous adjacency "x-x" indicates that the directionality is undecided due to conflicting orientation rules. The link color refers to the cross-MCI value, which indicates the strength of the relationships. The node color denotes the auto-MCI value, representing how strong the autocorrelation is.

## A.6 TERMS AND ABBREVIATIONS

Table 2: **Glossary of terms**

| Term | Definition |
| --- | --- |
| Causal discovery | Revealing causal information by analyzing purely observational data under certain assumptions. |
| Time trend | A function concerning time within a given data span. |
| Intrinsic trend | Time trends that are inherent to the fundamental mechanisms governing the variables (real trends). |
| Measurement trend | Time trends that are essentially observation errors unique to the recorded values (false trends). |
| Causal sufficiency | The absence of unobserved confounders. |
| Pseudo causal sufficiency | Any unmeasured confounding factors influencing the relationship interested can be adequately represented by a smooth mathematical function of time. This implies that the only unobserved confounders are those inherent in time trends. |
| Causal Markov condition | All the relevant probabilistic information that can be obtained from the system is contained in its direct causes, or, expressed differently, if two variables are not connected in the causal graph given some set of conditions, then they are conditionally independent. |
| Causal faithfulness | Independencies in data arise not from coincidence, but rather from causal structure or, expressed differently, if two variables are connected by a causal link in the graph. |

**Table 2 continued from previous page**

| Term | Definition |
|---|---|
| Leaf node | Nodes without any descendants. |
| Changing causal module | A component within a causal model or system where the causal relationships can change over time or across different contexts. This concept acknowledges that the dependence between variables are not static and can evolve due to various factors such as shifts in underlying mechanisms. |

**Table 3. Abbreviations**

| Abbreviation | Full description |
|---|---|
| PC algorithm | The Peter-Clark algorithm |
| SGS algorithm | The Spirtes-Glymour-Scheines algorithm |
| FGES algorithm | The Fast Greedy Equivalence Search algorithm |
| PCMCI+ algorithm | The Peter-Clark-momentary-conditional-independence plus algorithm |
| TrendDiff | Trend Differentiator |
| TIN | Transformed Independent Noise |
| SEM | Structural Equation Model |
| OICA | Over-complete independent component analysis |
| MCI | Momentary conditional independence |
| KCI test | Kernel-based conditional independence test |
| PA | Parent |
| HPC | High-performance computing |
| CO | Carbon monoxide |
| $PM_{2.5}$ | Particulate matter with aerodynamic diameter $\leq 2.5 \mu m$ |
| $NO_2$ | Nitrogen dioxide |
| $O_3$ | Ozone |
| Temp. | Temperature |
| Humid. | Relative humidity |

