# OpenReview forum: "TrendDiff: Decoupling Intrinsic and Measurement Trends for Enhanced Time Series Causal Discovery"
_ICLR.cc/2025/Conference — Submitted to ICLR 2025_

### Official Review · Reviewer_v1z9 · 2024-10-31

**Soundness:** 3
**Presentation:** 3
**Contribution:** 2
**Rating:** 5
**Confidence:** 4

**Summary:**

This paper addresses the need to differentiate between intrinsic trends, which are integral to the causal interpretation, and measurement trends, which are artifacts of the measurement process rather than the data-generating mechanism. It is clearly shown that misidentifying one for the other will lead to incorrect causal interpretations. The authors first show that a trend, whether intrinsic or measurement-related, can be identified. More importantly, it is then argued that if a trended variable is affected by either an intrinsic or measurement trend, then the type can be distinguished, based on appropriate conditional independence statements related to the underlying causal structure. The paper concludes with a variety of experiments over simulated and real data, which demonstrate that the proposed algorithm can differentiate between the two types of trends. Further analysis is carried out at the level of causal discovery, where it is shown that appropritate detrending improves the recovery of the causal graph.

**Strengths:**

The paper addresses an interesting and important problem in causal inference, and proposes an intuitive framework and solution, with strong empirical backing.

I appreciated the use of causal graph illustrations, as this helped convey the intuition for the distinction between the downstream effects of intrinsic and measurement trends.

The experiments were well designed. The simulated and real experiments featured both types of trends, and the positive effect on detrending just the measurement trend for causal discovery is a nice touch. I particularly liked the selected real dataset, given the significance of the likely measurement trend.

**Weaknesses:**

As pointed out by other reviewers, the novelty of the paper is lacking, since many fundamental aspects come from the CD-NOD paper.

I think that the paper should be more transparent about not being able to simultaneously handle intrinsic and measurement trends for the same variable. This is only discussed at the very end. I would recommend clarifying it as a model assumption within Section 3. In general, I think that the model assumptions can be stated more clearly.

Most trends in the experiments section are sinuosoidal. There is a lack of evaluation for non-periodic trends (ex. Growth or decay).

I found some notations/definitions ambigious:

Although reasonably intuitive, intrinsic and measurement trend should be defined more rigorously in Definitions 2-3, given the importance to the paper.

I think that the phrasing "X2 and X3 are not observable" (line 63) should be adjusted, since it can easily be misinterpreted to mean that they are completely unobserved, rather than their true values not being observed.

What is C precisely? Is it the set of measurement times (modeled as a r.v.)?

Figure 6a) legend typo "Guassian" -> "Gaussian"

**Questions:**

What would the causal discovery results look like if it was completely detrended, as is often done in the literature (intrinsic trends also removed)? It would be interesting to see this ablation if possible.

Do the authors think that distinction between measurement and intrinsic trends (given in this paper for a particular type of SEM) can be generalized when formalized in the SDE setting? There has been recent work with SDEs being used as SEMs (see for example from the last year: "Dynamic Structural Causal Models" and "Causal Modeling with Stationary Diffusions"). These models seem quite applicable, especially for modeling financial markets.

---

### Official Review · Reviewer_zCPg · 2024-11-03

**Soundness:** 2
**Presentation:** 2
**Contribution:** 1
**Rating:** 3
**Confidence:** 4

**Summary:**

This paper studies trends in causal discovery. Using a time indexing variable, the authors show how to identify variables with trend and how to differentiate intrinsic trend v.s. measurement trend variables. They also conduct experiments on synthetic data and real data to verify their theories.

**Strengths:**

Extensive experiments are conducted on both synthetic data and real data to verify the results.

**Weaknesses:**

**1**. The authors seem to confuse the concept of trend with nonstationarity. Nonstationarity means the causal mechanism $f_i$ that generates $X_i$ from its parent $PA_i$ changes with time. For trend, such a change should has a direction (increasing or decreasing) with time, see Def. 1 of (White and Granger, 2011) in your reference. However, Def. 1 in the paper fails to signify this.

In a similar way, measurement trend should not be confused with meansurement error along time. According to Def. 3 in the paper, a white noise along time or a Brownian motion can be considered as measurement trend, however, there is no tendency in them.

**2**. The novelty of the proposed techniques is limited. The use of a time indexing variable $C$ to identify nonstationary variables is a classic approach established by CD-NOD (Huang et al., 2020). I believe that trend warrants special attention, particularly regarding their properties of increasing or decreasing over time, rather than trivially applying results from nonstationary causal discovery.

**Questions:**

See weakness above.

---

### Official Review · Reviewer_qk2J · 2024-11-04

**Soundness:** 2
**Presentation:** 2
**Contribution:** 2
**Rating:** 3
**Confidence:** 5

**Summary:**

This paper introduces TrendDiff, an algorithm designed to enhance causal discovery in time-series data by distinguishing between intrinsic trends (genuine changes in the underlying phenomenon) and measurement trends (artifacts or errors in observation). Recognizing that measurement trends can obscure causal relationships, the authors present a two-phase approach: identifying trend-affected variables, and then classifying these into intrinsic or measurement trends based on their structural and conditional independence relationships within the causal network. The authors validate TrendDiff through simulations and a real-world environmental health dataset, showing that removing measurement trends improves causal discovery accuracy. By employing techniques like the “Savitzky-Golay filter” for detrending, they demonstrate that eliminating measurement noise clarifies causal structures, enabling more accurate data-driven insights.

**Strengths:**

- The introduction of the TrendDiff algorithm for distinguishing intrinsic and measurement trends in time-series data is innovative and addresses an important gap in causal discovery for observational datasets.
- The algorithm’s two-phase structure, employing conditional independence tests and Savitzky-Golay filtering, is an effective approach for identifying and removing measurement trends, demonstrating improved accuracy in causal inference over baseline methods.
- The work is relevant to fields like finance, environmental science, and healthcare, where noisy trends can obscure causal relationships, offering a potentially valuable tool for enhancing data pre-processing and improving causal analysis reliability in these domains.

**Weaknesses:**

- The current model assumes that all causal relationships are instantaneous, an oversimplification for domains requiring lagged causality (e.g., health impacts of pollution). To broaden its relevance, the algorithm could integrate methods for time-lagged effects allowing it to account for delayed causation in complex time-series data.
- A crucial element of the TrendDiff algorithm is Algorithm 2, which represents the main novelty of this paper. However, this algorithm is placed in the appendix rather than the main text, obscuring its importance. Upon closer examination, it appears that Algorithm 2, along with other components, primarily serves as a data preprocessing step rather than a standalone causal discovery technique. This distinction is not clearly stated in the paper, which may lead to confusion about the intended scope and purpose of the method. Meanwhile, Algorithm 1, which is nearly identical to the CD-NOD algorithm, could be moved to the appendix, as it offers limited novel contribution and primarily replicates an established process for detecting time-influenced variables. Additionally, a key part of this preprocessing involves the Savitzky-Golay filter to detrend measurement-influenced variables and reduce noise. However, the filter’s role, configuration, and the rationale for its selection are not discussed in the main paper, leaving a gap in transparency regarding this essential step. A clear justification for using the Savitzky-Golay filter, including details on parameterization and potential alternatives, would improve understanding of the algorithm’s robustness and clarify its intended application as a preprocessing technique.
- The paper does not provide a complete description of how the final causal graph is constructed, leaving out important details on whether additional steps were applied beyond the initial graph derived in Phase 1. While the initial causal graph is generated through conditional independence tests and orientation based on dependency relationships, it is unclear if this graph is directly used as the final causal graph or if further refinement steps were applied. The absence of a full algorithm or detailed methodology for finalizing the causal graph limits the reproducibility and clarity of the approach. Including this information would clarify whether the initial skeleton suffices or if other techniques (such as additional orientation rules or refinement methods) are necessary to achieve the final, fully directed causal graph.
- The proposed TrendDiff algorithm is based on CD-NOD, which theoretically identifies time-influenced variables as trends. However, an examination of the CD-NOD GitHub repository and example datasets (https://github.com/Biwei-Huang/Causal-Discovery-from-Nonstationary-Heterogeneous-Data/tree/master) indicates that the synthetic data relies on sinusoidal functions to represent time dependencies. Sinusoidal patterns more accurately model seasonality rather than long-term trends, introducing ambiguity in Phase 1 of TrendDiff. This means that Phase 1 of the algorithm may be detecting seasonality along with trends, which undermines the core objective of differentiating intrinsic trends from measurement errors. Without a clear separation between seasonality and true trends, the validity of the paper's approach is in question, as the methodology may not be isolating trends as intended. A more refined approach to differentiate between seasonality and trends would strengthen the algorithm’s theoretical foundation and improve its practical applicability.
- In Section 5.1, the authors state that "all trends were modeled as sinusoidal functions with periods w chosen randomly from a uniform distribution Unif([5, 25])". However, this approach models seasonality rather than a true trend, as sinusoidal functions are inherently cyclical and represent recurring patterns over time. True trends, in contrast, are typically non-repeating, directional changes that occur over a longer timeframe. By modeling trends with sinusoidal functions, the paper conflates seasonality with trend, which may lead to inaccuracies in trend differentiation and limit the algorithm’s effectiveness in real-world applications, where identifying genuine trends—rather than cyclic seasonal effects—is crucial.
- Phase 1 of TrendDiff relies on CD-NOD to generate the initial causal graph, using a surrogate variable to represent time. However, CD-NOD’s approach is nearly identical to the PC algorithm and does not account for time lags in causal relationships. Given that the proposed approach uses the causal skeleton and time-influenced variables as foundational components, employing algorithms like JPCMCI+ (https://proceedings.mlr.press/v216/gunther23a/gunther23a.pdf) and CDANS (https://proceedings.mlr.press/v219/ferdous23a/ferdous23a.pdf) would likely produce a more accurate causal skeleton with time lag considerations. Additionally, these algorithms are more efficient in handling high-dimensional data, which would improve the scalability and robustness of the method.
- The experimental section lacks a comprehensive evaluation of TrendDiff's performance relative to similar algorithms, such as JPCMCI+ and CDANS, both of which incorporate the time index as a surrogate variable and are better suited for time-series data. Benchmarking TrendDiff against these methods would provide a more robust assessment of its efficiency and accuracy, offering insights into its relative strengths and weaknesses. While the paper focuses solely on TrendDiff, the real data analysis only presents results from PCMCI+ with TrendDiff, without evaluating the effectiveness of TrendDiff or comparing it with other methods. Given that TrendDiff primarily functions as a data preprocessing step rather than a standalone causal discovery approach, a comparative study using different causal discovery methods on synthetic datasets, both before and after applying TrendDiff, would yield more meaningful insights. Such an evaluation would clarify the preprocessing step's impact on causal discovery accuracy and enhance understanding of its role in improving downstream analyses. Additionally, Figure 6(b), titled "Performance of PC algorithm using data pre- and post-elimination of detected measurement trends," compares the PC algorithm's performance before and after removing measurement trends. However, the PC algorithm is not suitable for time-series data as it assumes an i.i.d. structure and does not consider temporal lags, which prevents it from capturing the dependencies and causal lags inherent in time-series data. This limitation raises concerns about the validity of using the PC algorithm in this evaluation, as the lack of time-series-specific adaptations undermines the reliability of the results presented in Figure 6(b) and calls into question the overall evaluation method's accuracy in a time-dependent context.

**Questions:**

- Have you considered the possibility of variables being influenced by both intrinsic and measurement trends simultaneously? Could TrendDiff be adapted to detect and separate overlapping trends within a single variable, potentially using a hybrid model or disentanglement techniques?
- Could you clarify the choice of the Savitzky-Golay filter for detrending measurement trends? Including parameterization details and comparing this filter with other detrending methods would provide more insight into this choice and its impact on causal discovery.
- The construction of the final causal graph remains unclear. Was the initial causal graph sufficient, or did you apply additional refinement steps? A more comprehensive outline would improve the transparency of the final graph construction process.
- Despite the limitations of CD-NOD, why was it selected for Phase 1 instead of alternatives like JPCMCI+ or CDANs? Could you explain any advantages or the rationale behind this choice?
- Since CD-NOD uses sinusoidal functions, which represent seasonality rather than true trends, could Phase 1 of TrendDiff be misidentifying seasonality as trends?
- Why were sinusoidal functions chosen for trends in synthetic data, given their alignment with cyclical rather than directional changes? Clearly distinguishing seasonality from true trends could improve the approach’s validity, particularly in contexts with prevalent seasonal patterns.
- Why did you choose the PC algorithm to compare performance before and after eliminating measurement trends, given that PC is not designed for time-series data and cannot account for temporal dependencies? - Wouldn’t it have been more suitable to use CD-NOD, which was already employed in Algorithm 1 and is better suited for time-series causal discovery?

---

### Official Review · Reviewer_3DD7 · 2024-11-04

**Soundness:** 2
**Presentation:** 2
**Contribution:** 2
**Rating:** 5
**Confidence:** 3

**Summary:**

The authors present a constraint-based algorithm to detect all trend-influenced variables and partially distinguish intrinsic trends from measurement trends.

**Strengths:**

1. The proposed algorithm addresses an interesting topic in time series datasets. The algorithm is capable of detecting time trend variables, and given the definitions of intrinsic trends and measurement trends, it can further partially identify which time trend variables exhibit intrinsic trends and which are affected by measurement trends within certain causal structures.

2. The algorithm has been applied to a series of simulation experiments and a real case study.

3. The assumptions are stated in a comprehensive manner.

**Weaknesses:**

1. Without an explicit statement in the paper, this algorithm appears to be an extension of CD-NOD. The main paper describes phase 1 (Algorithm 1) in detail; however, I believe Algorithm 1 and Theorem 1 have already been introduced in Huang et al., 2020. The citation in phase 1 gave the impression that only step 4 in Algorithm 1 and the complete proof are originally presented in Huang et al., 2020. However, after reviewing the previous work by Huang et al., 2020, it seems the entire Algorithm 1 is from that paper. Therefore, I think the citation in this section is problematic.

2. There is no theorem guaranteeing phase 2, whereas Theorem 1 in phase 1 has been proved in previous work.

3. Considering that measurement trends are a type of measurement error, could the models introduced in the related work aimed at causal discovery in the presence of measurement errors be used as baselines? I understand that there are no previous methods capable of identifying time trend variables, but for the causal discovery part, these models could at least be treated as baselines in Fig. 6 (b). The effectiveness of the proposed algorithm for estimating the correct causal graph is difficult to evaluate without any baselines.

4. Algorithm 1 and Theorem 1 are demonstrated in a setting that assumes there are no measurement errors, correct? If there are errors in the dataset, how can it be guaranteed that the results from Algorithm 1 are correct, given that phase 2 heavily relies on the correctness of Algorithm 1?

**Questions:**

1. Are there any experimental results on the performance of identifying measurement-trend variables, similar to Fig. 6 (a)?

2. In Algorithm 2, measurement-trend variables are collected from the time-trend variables identified in Algorithm 1 after removing the identified intrinsic-trend variables. However, since intrinsic-trend variables are partly identified from the measurement, the remaining set of time-trend variables is not necessarily composed solely of measurement-trend variables, correct?

3. The same question as in Questions Item 2 applies to Figure 6 (b). If the 'measurement-trend' variables are a mixture of measurement- and intrinsic-trend variables, how can it be ensured that the experiments were conducted both before and after the elimination of 'detected measurement trends'?

4. Please direct me to the relevant part if I missed it—could you explain how the measurement trends are removed in the experiment?

5. Is there a reason for choosing the PC algorithm, which is designed for IID samples, after removing the measurement trends, instead of applying Algorithm 1 or other algorithms designed for time series datasets? If applying Algorithm 1 to the data after the elimination of detected measurement trends results in higher accuracy, does this imply that the existence of measurement trends indeed affect the correctness of Phase 1 and, consequently, the entire proposed algorithm (please refer to Weaknesses Item 4)? This seems to suggest a cyclic conflict.

---

### Meta-Review · Area_Chair_MdAZ · 2024-12-17

**Metareview:**

All reviewers mentioned several weaknesses and problems, and none of them assigned a positive score. Most importantly, the paper has been criticised because of its strong similarity to (Huang et al. 2020) and unclear novelty. Further,  part of the theoretical analysis seems to hold only in the limit of zero measurement errors, and the influence of noise remains unclear. There was no rebuttal, so these are still open questions. Therefore, I recommend rejection of this paper.

**Additional Comments On Reviewer Discussion:**

There are many open questions, but no rebuttal...

---

### Decision · Program_Chairs · 2025-01-22

Reject